# Production of D-Lactate from Avocado Seed Hydrolysates by Metabolically Engineered *Escherichia coli* JU15

**Dulce María Palmerín-Carreño, Ana Lilia Hernández-Orihuela and Agustino Martínez-Antonio \***

Departamento de Ingeniería Genética, Cinvestav Irapuato, Km. 9.6 libramiento norte Irapuato-León, Irapuato 36824, Mexico; dulce.palmerin@cinvestav.mx (D.M.P.-C.); lilia@biosintetica.mx (A.L.H.-O.)

\* Correspondence: agustino.martinez@cinvestav.mx; Tel.: +52-462-623-9660

**Abstract:** Agroindustry residues can be used to produce valuable chemicals such as lactic acid, which is a primary chemical platform with many industrial applications. Biotechnological processes are the main approach of lactic acid production; however, culture media has an important impact on their costs. As a result, researchers are exploring various methods of production that use residual or waste biomass as raw materials, most of which are rich in lignocellulose. Nevertheless, starch and micronutrients such as those contained in avocado seeds stand out as promising feedstock for the bioprocess as well. In this study, the lactogenic *Escherichia coli* strain JU15 was evaluated for producing D-lactate using an avocado seed hydrolysate medium in a controlled stirred-tank bioreactor. The highest lactic acid concentration achieved was 37.8 g L$^{-1}$ using 120 g L$^{-1}$ as the content of initial reducing sugars. The results showed that D-lactate can be produced from avocado seed, which hydrolysates to 0.52 g L$^{-1}$ h$^{-1}$ using the engineered *E. coli* JU15. This study may serve as a starting point to further develop bioprocesses for producing metabolites using avocado seed hydrolysates.

**Keywords:** D-lactate; *Escherichia coli*; avocado seed hydrolysate; stirred-tank bioreactor

## 1. Introduction

Residual agroindustry has two main groups: lignocellulosic and starch-enriched materials. The first one corresponds principally to crop residues, and the second one corresponds to tubercles and their fruit seeds, such as the avocado seed. Avocado is mainly produced in North and Central America; in México, avocado is a main agroindustry commodity exceeding a global production of 1.9 million metric tons (MMT) in 2018 [1]. The industrial processing of this fruit, mostly for guacamole and oil production, yields considerable amounts of residues such as the biomass constituted by husks and seeds; the latter accounts for up to 26% of the fruit fresh mass [2]. The composition of the lyophilized avocado peeled seeds (Hass variety) mostly consists of carbohydrates (48–79.5%) and other minimal components, among them: lipids (5.5–15%), proteins (3.4–19%), crude fiber (4.0%), ashes (0.8–4.2%), and moisture (13.2%) [2,3]. An ultimate analysis of seeds results in: total carbon (85.9%), hydrogen (0.55%), nitrogen (3.17%), oxygen (9.49%), sulfur (0.07%), volatile matter (27.55%), and biologically important minerals such as potassium, calcium, magnesium, iron, zinc, phosphorus, and silicon [3,4]. Starch represents nearly 60% of the seed (dry matter basis) [5], resulting in large amounts of potentially fermentable sugars. Consequently, avocado seeds stand out as promising feedstock for industrial fermentations and applications within the biorefinery concept.

In a recent review, lactic acid has been included in the list of the "Top 10" chemical opportunities for biorefinery after considering the following criteria: the extensive recent literature about the molecule, the multiple applicability of the product, the ability of this molecule to be produced at large

volumes, and its use as a building block molecule [6,7]. The study of lactic acid bacteria can be traced back to the beginning of the 1900s [8], and a few years later, its applications for cheese fabrication and silage started. The study of the enzymes that are responsible for lactic acid production was registered in 1938 [9], and its production via the use of recombinant microorganisms was reported by the end of the 20th century [10]. However, the use of waste material to produce lactic acid is relatively recent [11]. A wide range of bacteria and fungi are used to produce lactic acid, the most important being the genera *Lactobacillus*, *Lactococcus*, *Bacillus*, and *Enterococcus* for bacteria, and *Rhizopus*, *Aspergillus*, and yeast for fungi [12]. The most engineered microorganisms' models that have been used to produce lactic acid are *Escherichia coli*, *Bacillus subtilis*, *Pichia pastoris*, and *Saccharomyces cerevisiae*, among others [13].

For the production of lactate in bacteria and fungi, it has been common practice to use raw materials from agroindustry. Those include potato, sugarcane bagasse, and corncobs for fungi; and mixed food waste, sugarcane bagasse, mango seeds, orange peel, green peas, curcuma longa, chicory flour, wheat bran, spent distiller's and brewer's solids, and pulp mill residue for bacteria [14].

For microbial bioprocesses, the composition of the culture media is an important factor. It represents the source of molecular building blocks for growth and the production of metabolites. Furthermore, the quality and cost of the culture media impact the global efficiency and economy of biotechnological lactic acid production [8]. Indeed, a bottleneck cost is the substrate, in particular that of the carbon-source cost. Therefore, finding cheap and renewable feedstocks such as agroindustry residues represents a field of opportunity in this industry [15,16].

The biotechnological use of agricultural wastes and residues, such as avocado seeds, frequently requires prior processing of the natural resource in order to make nutrients available. On the other hand, the engineering of bacteria is needed in order to develop specific strains that are capable of metabolizing C6 sugar mixtures (glucose and lower quantities of mannose and galactose), and in some cases C5 (xylose) reducing sugar mixtures obtained by the more common hydrolysis processes of raw materials. In addition, these latter strains tolerate the potential inhibitors contained in the hydrolysates [17,18]. Also, to avoid cumbersome racemic mixture separations, the production of optically pure enantiomers, such as D-lactate or L-lactate, are required for several applications. Indeed, several metabolically engineered *E. coli* strains produce D-lactate and show advantages when using plant hydrolysates as feedstock [19–22]. Among such strains is JU15, which is able to convert xylose into D-lactate with high productivity and yield (0.8 gD-LA L$^{-1}$ h$^{-1}$ and 0.95 gD-LA gXYL$^{-1}$, respectively) when grown in mineral media [23].

Therefore, the aim of this study was to evaluate the use of avocado seed hydrolysate (ASH) as a feedstock resource to produce D-lactate using the metabolically engineered *E. coli* JU15 in a controlled 3-L bioreactor.

## 2. Materials and Methods

### 2.1. Microorganism and Growth Conditions

The *Escherichia coli* JU15 strain, generated in the Biotechnology Institute at National Autonomous University of Mexico (UNAM) México, has been previously used for *D*-lactic acid production using C6 and C5 sugars, corn stover, or sugar cane bagasse hydrolysates [23,24]. Here, we evaluated the capacity of this strain for D-lactic acid production using avocado seed hydrolysate as a substrate. The bacterial homolactogenic strain used in this work was the *E. coli* strain JU15 (*E. coli* MG1655: Δ*pflB*, Δ*adhE*, Δ*frdA*, Δ*xylFGH*, *gatC-S184L*, Δ*reg 27.3 kb*$^r$, *km*$^r$) [23,24]. Starting from an overnight inoculation of *E. coli* JU15 on LB solid Petri dish at 37 °C, one isolated colony was cultivated overnight at 37 °C in 30 mL of LB medium [25] in a 250-mL baffled flask at 180 rpm. This culture was centrifuged for 20 min at 3000 rpm and 4 °C (DuPont Sorvall RC-5B, Wilmington, DE, USA), and the cell pellet was washed twice with an equal volume of M9 5X salts (see below) and restored to their original volume with these same M9 5X salts. Several aliquots of this culture were maintained as glycerol (15%) stock and stored at −80 °C. Pre-inoculum was prepared by inoculating 1 mL of JU15 stock glycerol in a 250-mL baffled

flask containing LB medium (30 mL) and incubated overnight at 37 °C at 180 rpm, to be used as stock inoculum. In JU15 cultures, the antibiotic kanamycin was added to 30 μg mL$^{-1}$.

### 2.2. Avocado Seed Hydrolysate Medium (ASH) Preparation

Avocado (*Persea americana* Mill) seeds were washed, broken into smaller fragments using a hammer mill, and dried in an oven (PRONALAB, Tlalnepantla, México) overnight at 60 °C. Subsequently, seeds fragments were finely ground in a ball pulverizing mill, and the powder was stored in black bags in a cold place to avoid humidity. When required, a quantity of avocado seeds powder was hydrolyzed with hydrochloric acid (HCl) (1.0% *v/v*) under a thermal pressure treatment (120 °C, 0.1 MPa) for 15 min as described by Tzintzun-Camacho et al. [26] (2016) and the patent application WO/2016/079568 [27]. The amount of seeds powder per liter of hydrolysate ranged from 10% to 30% w v$^{-1}$, depending on the desired final concentration of reducing sugars. For the preparation of the experimental medium, one volume of concentrate hydrolysate was added with 5× salts of M9 minimal medium, and a final volume with deionized water before sterilization. The concentration of 5× salts in the final medium were: 7 g L$^{-1}$ Na$_2$HPO$_4$7H$_2$O, 3 g L$^{-1}$ KH$_2$PO$_4$, 1.0 g L$^{-1}$ NH$_4$Cl, and 0.5 g L$^{-1}$ NaCl.

### 2.3. Bioreactor Cultivation

A 500-mL baffled flask containing ASH medium (100 mL) was inoculated with 10 mL of the JU15 stock inoculum and incubated at 37 °C at 180 rpm for 15 h. This was used as bioreactor inoculum. The fed-batch cultivations were carried out in a 3-L stirred-tank bioreactor (1-L working volume) (Applikon, Delft, The Netherlands) for 120 h in ASH medium containing 40 g L$^{-1}$, 70 g L$^{-1}$, and 120 g L$^{-1}$ of reducing sugars, which were inoculated with a 10% v v$^{-1}$ inoculum size.

The fed-batch experiments were carried out in a 500-mL initial working volume reactor (110 g L$^{-1}$ initial reducing sugars) with 10% v v$^{-1}$ inoculum size. Since the beginning of the experiment, 500 mL (40 g L$^{-1}$ reducing sugars) was started to feed the reactor (3 mL min$^{-1}$).

All of the bioreactor fermentations were maintained with 3-vvm aeration, 37 °C, 200 rpm, and pH values were kept between 6.6–7.0. All of the fermentations started with the equivalent of 0.15 g DCW L$^{-1}$. The temperature and stirring speed were controlled by using one six-bladed Rushton-type impeller; pH was controlled by the automatic addition of KOH (4 N). Three-mL samples of bacterial cultures were monitored for culture viability, biomass, D-lactate, and reducing sugars at 0 h, 16 h, 20 h, 24 h, 52 h, and 124 h (batch) and 0 h, 4 h, 8 h, 24 h, 28 h, 32 h, 48 h, 52 h, 56 h, and 72 h (fed-batch), culture viability (CFU mL$^{-1}$) was determined by plate count. All of the experiments were carried out in triplicate, and the average and standard deviation values of independent experiments are shown in the respective figures and tables.

### 2.4. Chemicals

The following chemicals were used in this study: lactic acid (85% purity, Sigma-Aldrich, St Louis, MO, USA), kanamycin (Sigma-Aldrich, St Louis, MO, USA), hydrochloric acid (Karal, León Gto, México), sulfuric acid (Karal, León Gto, México), potassium hydroxide (Karal, León Gto, México), dextrose (Sigma-Aldrich, St Louis, MO, USA), dinitro salicylic acid (Sigma-Aldrich, St Louis, MO, USA), peptone from casein (Merck Millipore, Burlington, MA, USA), yeast extract (Becton Dickinson, Franklin Lakes, NJ, USA), polysorbate (Tween 80) (Sigma-Aldrich, St Louis, MO, USA), dipotassium phosphate (Sigma-Aldrich, St Louis, MO, USA), sodium acetate (JT Baker, Phillipsburg, NJ, USA), ammonium citrate (Sigma-Aldrich, St Louis, MO, USA), magnesium sulfate (Sigma-Aldrich, St Louis, MO, USA), and manganese sulfate (Sigma-Aldrich, St Louis, MO, USA).

### 2.5. Analytical Methods

The optical density (OD) at 600 nm, which was measured using a Genesys 10S UV–Vis spectrophotometer (Thermo Scientific, Waltham, MA, USA), was used to estimate the bacterial growth in LB medium [28]. Inoculum was added to adjust the initial OD to ~0.2 units.

Bacterial cell biomass was quantified as dry cell weight (DCW). Samples were centrifuged at 3000 rpm for 20 min at 4 °C (DuPont Sorvall RC-5B, Wilmington, DE, USA), and the cellular pellets were dried in an oven (PRONALAB, Tlalnepantla, México) at 80 °C for 24 h. Weight was determined by gravimetry.

Lactic acid, acetic acid, and furans (furfural and hydroxymethylfurfural) was determined by HPLC (Agilent 1200, Santa Clara, CA, USA) using an aminex HPLC-87H column (Bio-Rad, Richmond, CA, USA) under the following conditions: 5 mM $H_2SO_4$ as the mobile phase at 0.8 mL $min^{-1}$ flow and 50 °C.

Reducing sugars present in the hydrolysate and in experimental culture samples were measured using the dinitro salicylic acid (DNS) method modified from Miller [29]. The DNS reagent was prepared as described by Saqib and Whitney [30].

The specific growth rate ($\mu_{máx}$) was calculated considering the exponential growth phase for each experiment. The product/substrate yield (YD-LA $Sugars^{-1}$) was calculated taking into account the whole fermentation time for each experiment and, for a fair of comparison, the volumetric productivity of D-lactate (QD-LA) was also calculated at 48 h of cultivation.

## 3. Results

### 3.1. Biomass Production of Lactogenic E. coli JU15 as a Function of the Quantity of Initial Reducing Sugars in 3-L Bioreactor

In a previous study [26], it has been demonstrated that ASH supplemented with 5× M9 minimal medium salts was able to support the growth of *E. coli* even better than LB medium. Since the ASH medium can be considered natural and complex, by its plant seed origin, it is very difficult to know all their components. As mentioned, the main components in the seed are carbohydrates mainly stored in starch; for a practical purpose, it was measured as reducing sugars in the hydrolysate. Since that carbon source is the main limiting factor for bacterial growth, the value of reducing sugars was taken into account to decide when to use different quantities (volumes) of ASH for preparing the experimental medium. Then, the performance of the ASH was tested for growing the recombinant lactogenic *E. coli* JU15. For these assays, we used a range from 40 g $L^{-1}$ (obtained from the hydrolysis of 10% avocado seed powder) to 120 g $L^{-1}$ (30% avocado seed powder) of reducing sugars for preparing culture medium supplemented with 5× M9 minimal medium salts. Due the components of the ASH medium, it has a red–brown color, and therefore it was difficult to follow the bacterial growth by turbidometry. Hence, DCW biomass was obtained from culture samples (Table 1) and colony-forming units (CFU) were obtained from the final culture (Table 2), the maximum biomass production was 2.73 ± 0.18 gDCW $L^{-1}$ at 120 g $L^{-1}$ of initial reducing sugars (Table 1). For comparison, the wild-type *E. coli* K12 MG1655 strain reach a maximal biomass of 2.25 gDCW $L^{-1}$ at 20 g $L^{-1}$ of reducing sugars [26].

**Table 1.** Results of different fermentation processes, varying the initial reducing sugar concentration.

| System | Initial Reducing Sugars (g $L^{-1}$) | Residual Reducing Sugars (g $L^{-1}$) | Lactic Acid Production (g $L^{-1}$) [†] | Final Biomass (g $L^{-1}$) | $Y_{D-LA}$ ($g_{D-LA}$ $g_{sugars}^{-1}$) | Overall Volumetric Productivity (mg $L^{-1}$ $h^{-1}$) |
|---|---|---|---|---|---|---|
| Batch | 40 ± 1.8 | 21.9 ± 0.8 | 17.9 ± 0.5 | 2.22 ± 0.15 | 0.42 ± 0.05 | 0.25 ± 0.06 |
| Batch | 70 ± 0.4 | 50.7 ± 1.3 | 18.9 ± 0.2 | 2.57 ± 0.07 | 0.26 ± 0.10 | 0.26 ± 0.01 |
| Batch | 120 ± 3.3 | 83.1 ± 0.2 | 37.8 ± 0.3 | 2.73 ± 0.18 | 0.31 ± 0.08 | 0.52 ± 0.07 |
| Fed-Batch | 110 ± 4.1 (initial) | 21.4 ± 0.7 | 37.6 ± 0.4 | 2.52 ± 0.05 | 0.33 ± 0.07 | 0.48 ± 0.05 |

[†] after 48-h fermentation.

### 3.2. Production of D-LA and Consumption of Reducing Sugars by the Lactogenic E. coli JU15 in Batch Fermentations Using the ASH Medium

Besides measuring biomass, the consumption of reducing sugars and the production of D-LA on each batch bioreactor kinetics of *E. coli* JU15 were also quantified. The samples growing on the different preparations of ASH medium with starting quantities of reducing sugars at 40 g L$^{-1}$, 70 g L$^{-1}$, and 120 g L$^{-1}$ (Table 1 and Supplementary Materials) were assayed. At 40 g L$^{-1}$, the lactogenic bacteria consumed 48% of reducing sugars in 72 h and produced 18 g L$^{-1}$ of lactic acid after 48 h of fermentation. There was not a significant difference in lactic acid production between 48–72 h of the bioprocess. During the growth kinetics, it was observed that *E. coli* reaches the maximum dry biomass production in 8 to 10 h; from this time onward, the stationary phase was present. When increasing the quantity of initial reducing sugars to 70 g L$^{-1}$, we observed that *E. coli* slightly increase the production of lactic acid to 19 g L$^{-1}$ at 48 h, but the consumption of reducing sugars was just 40%. When the initial reducing sugar concentration was further increased up to 120 g L$^{-1}$, the amount of reducing sugars consumed was barely 35%; however, the lactic acid production increased to 37.8 g L$^{-1}$ (Table 1). The kinetic parameters for the fermentation at 120 g L$^{-1}$ of reducing sugars is shown in Table 2, showing the maximum number of cells (as colony-forming units: CFU), the specific growth rate (μ), D-LA volumetric productivity (at 123 h), and D-LA yield on reducing sugars. The specific growth rate and the doubling time were 0.63 h$^{-1}$ and 1.1 h, respectively, being very close values to those reported for *E. coli* strains growing in LB medium in aerobic conditions also (0.4 h$^{-1}$, 0.20 h) [18]. The viable cell count was $5 \times 10^{10}$ CFU mL$^{-1}$ after 48 h of fermentation. The maximum lactate concentration was $37.8 \pm 0.3$ g L$^{-1}$, which was achieved after 123 h of fermentation. The average yield of the product to substrate (lactic acid on reducing sugars) was 0.31 g g$^{-1}$ (Figure 1, and Supplementary Materials).

**Table 2.** Batch bioreactor kinetic results with 120 g L$^{-1}$ of initial reducing sugars.

| Parameter | Bioreactor Results |
|---|---|
| $X_{max}$ (CFU mL$^{-1}$) | $5.0 \times 10^{10}$ |
| Specific growth rate, $\mu$ (h$^{-1}$) | 0.63 (R-sq = 0.97) |
| Doubling time, $t_d$ (h) | 1.1 |
| $Q_{D-LA}$, (gD-LA L$^{-1}$ h$^{-1}$) | $0.31 \pm 0.08$ |
| $Y_{D-LA}$, (gD-LA g$^{-1}$ reducing sugars) | $0.31 \pm 0.08$ |
| Reducing sugar consumption (%) | $34.7 \pm 1.2$ |

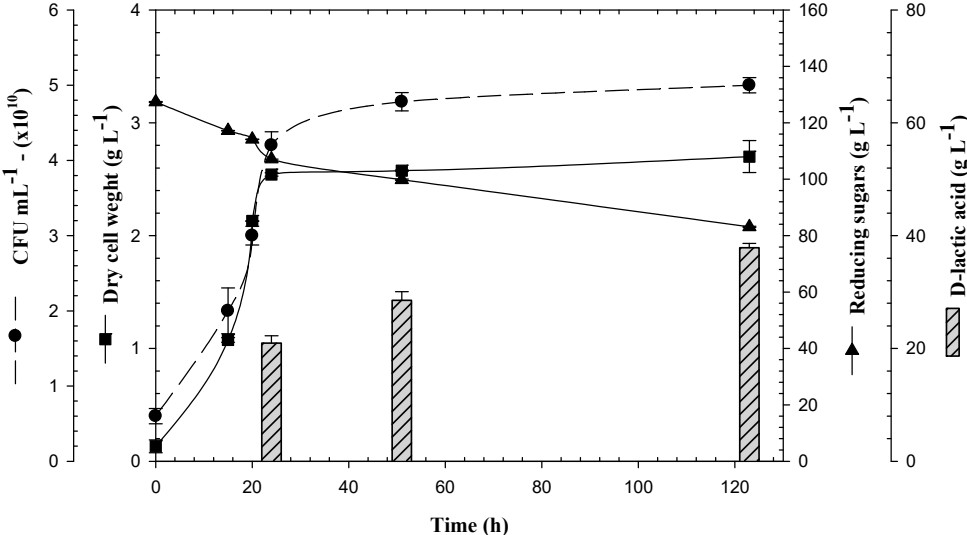

**Figure 1.** Kinetic of *Escherichia coli* JU15 in batch fermentations at 120 g L$^{-1}$ initial reducing sugars, 200 rpm, 37 °C, and pH 6.6. Data from two independent experiments are shown.

### 3.3. Feedback Fermentation Slightly Improves the Productivity of Lactic Acid by E. coli JU15 in the ASH Medium

In the scientific literature, lactic acid production has been tested using different fermentation techniques such as batch, fed-batch, and continuous cultures. The highest lactic acid concentrations have been reported to be achieved through batch and fed-batch fermentations, while higher productivities have been obtained in continuous cultures. The latter fermentation processes have the additional advantage of the production lasting for longer periods of time [31]. Therefore, it was interesting to further explore fed-batch fermentation. For this, the culture was initiated with a middle volume (500 mL) at 110 g L$^{-1}$ of reducing sugars, and the bioreactor was fed from the beginning with fresh medium (3 mL/min$^{-1}$) containing an equivalent of 40 g L$^{-1}$ of reducing sugars. With this fed-batch strategy, the average yield of the product to substrate (lactic acid on reducing sugars) slightly increased to 0.33 g g$^{-1}$, but produced around the same quantity of lactic acid (38 g L$^{-1}$) as in the last batch experiment. However, it is important to note that with the conditions of fed-batch assayed here, the bioprocess did not improve for biomass nor for lactic acid production (Figure 2). It should be interesting to further explore the bioprocess, changing parameters such as the rate of aeration, nutrients supply to ASH, etc., in order to further optimize the production of D-LA using ASH as a natural resource.

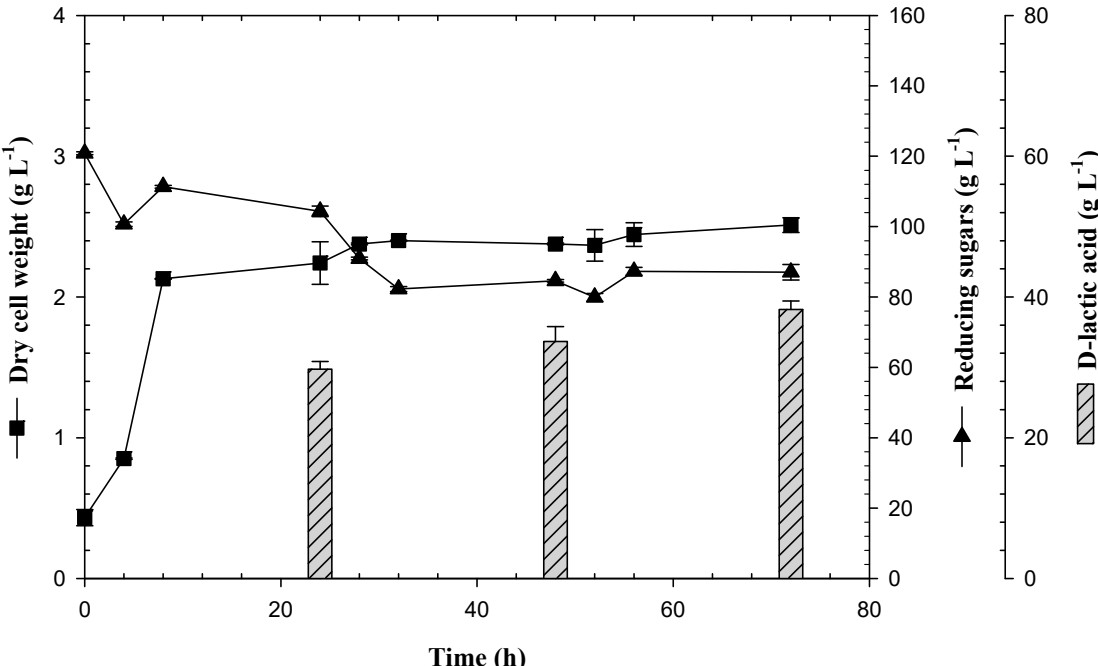

**Figure 2.** Kinetic of *Escherichia coli* JU15 in a fed-batch bioreactor at 110 g L$^{-1}$ initial reducing sugars, 200 rpm, 37 °C, and pH 6.6. The arrow indicates the time of starting feed with fresh medium (40 g L$^{-1}$ reducing sugars). Data from two independent experiments are shown.

## 4. Discussion

It is seen that the *E. coli* strain JU15 is capable of using the nutrients that are available following the thermochemical hydrolysis of avocado seeds for growth and lactic acid production, the latter reaching values near 38 g L$^{-1}$. However, due the natural complexity of the ASH hydrolysate, it is difficult to know which nutrients are specifically being employed in the bioprocess, and whether some of its components have a negative impact on growth. For simplicity, the carbon source is tracked in the medium, measuring them as reducing sugars. However, given that reducing sugars do not necessarily correspond to fermentable sugars, alternative methods for quantification should be needed; one example is monitoring sugars such as glucose, which is specifically taken up by *E. coli*. Here might

be a scope of opportunity to engineer *E. coli* strains that are optimal for the use of those sugars present in the avocado seed hydrolysate. The JU15 strain used here was specifically engineered to not produce alternative fermentable metabolites to lactic acid. In addition, the genes of the xylose transporter that consume ATP in the process of introducing xylose into the cell have also been deleted in this strain.

The results obtained here are similar to others that have been reported in the literature, even though the medium composition and the carbon source were different. For instance, Utrilla et al. [24] cultivated the *E. coli* strain JU15 using a culture medium based on sugar cane bagasse hemicellulosic hydrolysate (40 g $L^{-1}$ glucose) and reported $5.9 \times 10^{10}$ CFU $mL^{-1}$ after 48 h of fermentation at 37 °C. After 72 h of fermentation, 80% of the initial sugars content remained. Then, the remainder of residual components in the hydrolysates of natural resources seems to be a recurrent theme. Other authors have grown *E. coli* using glucose as the only carbon source (50 g $L^{-1}$) in a continuous bioreactor, resulting in a maximum specific growth rate ($\mu_{max}$) of 0.51 $h^{-1}$ [32].

In this study, the production of lactic acid does not seem to be dependent on the concentration of cells in the reactor; as observed in the four treatments, a dry biomass concentration between 2.2–2.7 g $L^{-1}$ was reached in all of the cases independently of the initial concentration of reducing sugars. Some researchers [33], who worked with recombinant *E. coli* strains for lactic acid production in a bioreactor, reached 2.2 g $L^{-1}$ of dry biomass with 100 g $L^{-1}$ of xylose or glucose as the carbon source; in addition, they obtained concentrations up to 70 g $L^{-1}$ of lactic acid during 48 h of process with a pH of 6.7 and 35 °C.

Overall, the results presented in this study showed a lower production of lactic acid as compared to the best results reported in the literature [13,14]. We speculate that the possible causes may be the lack of important nutrients in the avocado seed hydrolysate, or alternatively, that some of their components could be inhibitory components for *E. coli* metabolism in this experimental medium. However, with respect to the latter, most of the antimicrobial compounds reported in avocado seeds are extracted with organic solvents at moderate temperatures [34,35], and presumably, the acid–temperature–pressure treatment could be destroying these antimicrobial compounds. It is worth mentioning that the seed avocado hydrolysates used in this study do not contain furfural, because xylose was not detected, and the xylan content in the avocado seed has not been reported. Also, since we failed to detect hydroxymethylfurfural, that suggests that the hydrolysis conditions employed did not promote glucose dehydration to this furan. Although we detected 0.9 g $L^{-1}$ of acetate in the experimental medium, it has been reported that strain JU15 is tolerant to higher concentrations of acetate [24]. So, one perspective to increase the lactic acid production in the ASH might be to work to elucidate the main relevant components in the hydrolysate and to find a medium formulation that improves the production of this metabolite in the bioprocess. The present work is the first of such an approach, which aims to explore the lactic acid production using the hydrolysate of avocado seeds, which is an abundant industrial residue in different countries such as México, and may serve as the basis for the further improvement and optimization of the bioprocess and for the sustainable production of this and other important industrial metabolites.

**Supplementary Materials:** The following are available online at http://www.mdpi.com/2311-5637/5/1/26/s1, Figure S1: Kinetic of *Escherichia coli* JU15 in batch fermentations at 40 g $L^{-1}$ initial reducing sugars, 200 rpm, 37 °C, and pH 6.6, Figure S2: Kinetic of *Escherichia coli* JU15 in bioreactor at 70 g $L^{-1}$ initial reducing sugars, 200 rpm, 37 °C, and pH 6.6., Figure S3: Photography of the bioreactor before and after the fermentation, Figure S4: HPLC profiles on D-lactate production.

**Author Contributions:** Conceptualization, A.M.-A.; methodology, experimentation and analysis, D.M.P.-C.; resources, A.L.H.-O.; writing—original draft preparation, D.M.P.-C.; writing—review and editing, A.M.-A.

**Funding:** This research was funded by FIT-SE-CONACYT (México), grant number 275912.

**Acknowledgments:** The authors thank Estefanía Sierra Ibarra, Alfredo Martinez, and Marcos Acosta Fernandez for their assistance with the fermentation experiments and critical reading of the manuscript, Yolanda Rodríguez and Berenice Cueva for their help with the HPLC quantifications, and Gertrud Lund and Abhishek Dutta for English editing.

**Conflicts of Interest:** The authors declare no conflict of interest. The funders had no role in the design of the study; in the collection, analyses, or interpretation of data; in the writing of the manuscript, or in the decision to publish the results.

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
