# Peer review of "Production of d-Lactate from Avocado Seed Hydrolysates by Metabolically Engineered Escherichia coli JU15"

_fermentation, doi:10.3390/fermentation5010026_

Round 1

Reviewer 1 Report

The article entitled "Production of D-lactate avocado seed hydrolysates by metabolically engineered Escherichia coli JU15" is interesting and well describe the main issue. In Introduction section Authors provide information which help create good backgroud to undertaken scientific problem. Materials and methods are good described and give a chance to repeate all this experiments. Results are describe in a good and understanding way. Conclusions are good summarization of all scientific goal and obtained results.

I recommend this manuscript for publication in Fermentation.

Below please find my minor comments:

- Abstract- Authors should not use abbreviation "D-LA" in this place and in the rest of the text without explanation what does in means first.

-literature position [1] - I have some concerns if this source of data is "scientific" enough

- line 46 - I am not agree that "for microbial bioprocess, the COMPOSITION of the culture media is a CRUTIAL FACTOR..." I thing that as much important are also other factors like aeration, temperature and of course acidity, which can be a limiting factor especially in this types of fermentations.

- lines 59, please provide first the ful name of sth. plus abrevuation and atfer this use the abbreviation (gD-LA/Lh or gD-LA/gXYL)

-Materials and methods - please provide the origin institution from E. coli was taken

-line 103 please provide the specific time of "regular intervals"

- line 202, please provide "E. coli" before "JU15".

Author Response

Reviewer 1

The article entitled "Production of D-lactate avocado seed hydrolysates by metabolically engineered Escherichia coli JU15" is interesting and well describe the main issue. In Introduction section Authors provide information which help create good backgroud to undertaken scientific problem. Materials and methods are good described and give a chance to repeate all this experiments. Results are describe in a good and understanding way. Conclusions are good summarization of all scientific goal and obtained results.

I recommend this manuscript for publication in Fermentation.

Below please find my minor comments:

- Abstract- Authors should not use abbreviation "D-LA" in this place and in the rest of the text without explanation what does in means first.

R; the abbreviation was substituted for the complete word.

-literature position [1] - I have some concerns if this source of data is "scientific" enough.

R; we actualize to a USDA reference from 2018, we prefer conserve this reference since refers to the economical context around the theme of this research.

- line 46 - I am not agree that "for microbial bioprocess, the COMPOSITION of the culture media is a CRUTIAL FACTOR..." I thing that as much important are also other factors like aeration, temperature and of course acidity, which can be a limiting factor especially in this types of fermentations.

R: we agree, we chance the word “crucial” for “important” and complete the sentence as; … important factor. It represents the source of molecular building blocks for growth and metabolites production. (line 59)

- lines 59, please provide first the full name of sth. plus abbreviation and after this use the abbreviation (gD-LA/Lh or gD-LA/gXYL)

R: OK, we did it.

-Materials and methods - please provide the origin institution from E. coli was taken

R: now it is mentioned of line 82

-line 103 please provide the specific time of "regular intervals"

R: the times of sampling are now specified (line 122)

- line 202, please provide "E. coli" before "JU15".

R: was done, thank you.

Reviewer 2 Report

In the opinion of many scientists is the only alternative biomass feedstock coal, which must compete with non-renewable raw materials. Biomass, in comparison to petrochemical raw materials, contains in its composition more oxygen, while less hydrogen and carbon. Considering the quantitative differences in the chemical composition of these two classes of raw materials, it seems that a larger group of chemical products can be obtained in lignocellulose biorefineries than from petrochemical raw materials. One of the platform compounds that can be obtained by biological methods from biomass is lactic acid. Receiving D-lactate from post-production waste from avocado fruits is a reviewed article.The subject matter of the publication is interesting, but the document requires a few changes before publication in the Fermentation journal:

1. Some words are used in strange contexts, e.g. line 20 is "was assayed", or rather the phrase "was tested" or "was investigated" is used more often. The same applies to "minor / major quantities". I know that the meaning is sustained, but it seems to me that "low / high quantities" are more often seen in chemical texts.

2. There is not even a paragraph of the text in the Introduction on lactic bacteria commonly used for the synthesis of lactic acid. Please, complete the work with this issue.

3. There is no comparison of avocado waste compositions with other agricultural wastes used as a source of fermentation digests. Please complete this issue.

4. The acid hydrolysis of lignocellulosic biomass causes the formation of fermentation inhibitors, for example, furfurals are formed from pentosans, etc. Are these compounds present in processes with HCl hydrolysates? Has the furfurali quantification been carried out? Have you studied the inhibitory effect of such compounds on the growth of microorganisms and the productivity of D-lactates? Please discuss this issue with literature data.

5. In the whole text there is no consistency in writing units. For example, we have 120 oC (line 84) and 37oC (line 71); 120 g L-1 (line 95) and 120 gL-1 (line 149). The same are inconsistencies in the literature footnotes. I refer the authors to the instructions with the "Instructions for Authors".

6. In Figures 1 and 2, the presented lines are only a combination of points, not curves, which can be described by the equation of the curve. These lines have a very strange course. I suggest correcting these drawings and either removing these lines or replacing them with appropriate curves.

I believe that the work can be published after minor revision.

Author Response

Reviewer 2.

In the opinion of many scientists is the only alternative biomass feedstock coal, which must compete with non-renewable raw materials. Biomass, in comparison to petrochemical raw materials, contains in its composition more oxygen, while less hydrogen and carbon. Considering the quantitative differences in the chemical composition of these two classes of raw materials, it seems that a larger group of chemical products can be obtained in lignocellulose biorefineries than from petrochemical raw materials. One of the platform compounds that can be obtained by biological methods from biomass is lactic acid. Receiving D-lactate from post-production waste from avocado fruits is a reviewed article. The subject matter of the publication is interesting, but the document requires a few changes before publication in the Fermentation journal:

1. Some words are used in strange contexts, e.g. line 20 is "was assayed", or rather the phrase "was tested" or "was investigated" is used more often. The same applies to "minor / major quantities". I know that the meaning is sustained, but it seems to me that "low / high quantities" are more often seen in chemical texts.

R: Thank your observations, we carefully corrected these words along the manuscript. Two colleagues help us with the English edition.

2. There is not even a paragraph of the text in the Introduction on lactic bacteria commonly used for the synthesis of lactic acid. Please, complete the work with this issue.

R: two paragraphs and 4 references were added on in the introduction section in the manuscript, lines 45 to 54.

3. There is no comparison of avocado waste compositions with other agricultural wastes used as a source of fermentation digests. Please complete this issue.

R: It is a very interesting suggestion, but a detail comparison could take a long of space. In their place we introduce a brief sentence at the beginning of the introduction section (line 27s) and a paragraph on line 55-58, in the same section.

4. The acid hydrolysis of lignocellulosic biomass causes the formation of fermentation inhibitors, for example, furfurals are formed from pentosans, etc. Are these compounds present in processes with HCl hydrolysates? Has the furfurali quantification been carried out? Have you studied the inhibitory effect of such compounds on the growth of microorganisms and the productivity of D-lactates? Please discuss this issue with literature data.

R: this is an important issue, thank you. In our analysis of the avocado hydrolysate we did not find furfural neither its derivatives. On the other hand, some studies were report the presence of antimicrobial compounds in avocado seeds, these long chains aliphatic acids are extracted with organic solvents at cold temperatures. We think our relatively aggressive acid and high temperature and pressure are enough to destroy these compounds but certainly it is pending to verify it, a sentence is introduced on the discussion section mentioning this point (line 254-260).   

5. In the whole text there is no consistency in writing units. For example, we have 120 oC (line 84) and 37oC (line 71); 120 g L-1 (line 95) and 120 gL-1 (line 149). The same are inconsistencies in the literature footnotes. I refer the authors to the instructions with the "Instructions for Authors".

R: thank you, these inconsistences were corrected along the manuscript

6. In Figures 1 and 2, the presented lines are only a combination of points, not curves, which can be described by the equation of the curve. These lines have a very strange course. I suggest correcting these drawings and either removing these lines or replacing them with appropriate curves.

R: thank you, we correct to line tendencies instead of curves in both figures.

I believe that the work can be published after minor revision.

Reviewer 3 Report

The present work is dealing with the lactic acid production by an engineered E. coli strain on avocado seeds-extract medium. The topic is interesting, and the work well designed and performed.  In my opinion, authors might add more details on E. coli engineering.  Given the low production of lactic acid, is the process based on avocado seeds extract in any case economically advantageous?

Overall, I recommend the publication of the manuscript.   

Author Response

Reviewer 3.

The present work is dealing with the lactic acid production by an engineered E. coli strain on avocado seeds-extract medium. The topic is interesting, and the work well designed and performed.  In my opinion, authors might add more details on E. coli engineering.  

R1: the genotype of the bacterium is mentioned on lines 85-87.

Given the low production of lactic acid, is the process based on avocado seeds extract in any case economically advantageous?

R2: At this point, we have not been made a techno-economical run about the process because we think we can improve it. On posterior intents to improve the production we can do it. Thank you.

Overall, I recommend the publication of the manuscript.   

R: thank you very much.